## Research Article

(2024). Snake words in Estonia: Language,
nature and extinction in Andrus Kivirähk's *The
Man Who Spoke Snakish*. *Cambridge Prisms:
Extinction*, **2**, e3, 1–6

colonialism; cultural extinction; decline; human
activities; indigenous

**Corresponding author:**
Alfie Howard;
Email: pt17ajh@leeds.ac.uk

*This article was updated on 20 April 2024.

# Snake words in Estonia: Language, nature and extinction in Andrus Kivirähk's *The Man Who Spoke Snakish**

Alfie Howard[1] and Diane Nelson[2]

[1]School of English, University of Leeds, Leeds, UK and [2]School of Languages, Cultures and Societies, University of Leeds, Leeds, UK

## Abstract

This article discusses Estonian author Andrus Kivirähk's novel *The Man Who Spoke Snakish* in the context of language extinction and biocultural diversity. The novel is set in Medieval Estonia, but the viewpoint of the protagonist as a speaker of a vanishing language from a vanishing culture resonates with the lived experience of millions of people who have lost lifeways and livelihoods to colonisation and cultural assimilation. The fictitious language of Snakish allows its speakers to integrate fully into the natural world and to form complex interdependent relationships with non-human animals. This web of nature, culture and language is destroyed by a colonising society that is anthropocentric, ecologically destructive and socially hierarchical, and which views nature as something to exploit or fear. The novel explores the emotions of grief and loss for both a culture and an ecosystem heading for extinction.

## Impact statement

This article uses the context of language extinction and biocultural diversity in the real world to approach a work of fantastical fiction, bridging the disciplinary boundary between literature and sociolinguistics in the context of extinction studies.

## Article

*The Man Who Spoke Snakish* is a novel by Andrus Kivirähk, first published in Estonian as *Mees, kes teadis ussisõnu* in 2007. Set in thirteenth-century Estonia during the Northern Crusades, the novel tells the story of Leemet, a speaker of an ancient, magical language, who is witnessing the steady decline of his culture, language and way of life. In the past, Leemet's family and their fellow Estonians lived in the forest, hunting animals, gathering berries and talking with snakes. With the arrival of the Crusaders, however, Estonians are moving to Christian villages to plough the land, serve foreign invaders and in the process, forgetting the ancient language of snakes.

In this article, we discuss Kivirähk's novel in the context of language extinction and biocultural diversity. The fictional, magical language of Snakish embeds the culture of Leemet and his fellow speakers as part of the natural world, in stark contrast to the anthropocentric outlook of the Crusaders and Christian villagers. As the Snakish language dies out over the course of the novel, so too does this grounding of the Estonian people within the forest ecosystem. While Kivirähk's narrative satirises Estonian nationalism and mythmaking, in depicting the decline of Snakish the novel expresses grief for the loss of a way of life while channelling contemporary anxieties about the future of the Estonian language.

### Language extinction in the real world

Alongside the extinction crisis for species and ecosystems, a parallel crisis for diversity is unfolding for the world's languages, with 50%–90% expected to disappear within the current century (Krauss, 1992). According to the language database Ethnologue, there are, as of February 2023, 7,168 languages spoken in the world. However, the distribution is wildly uneven, with some far more widely spoken than others: the 200 most widely spoken languages in the world, for example, are spoken as a native language by over 88% of the world's population, with many more people (several hundred million, according to Ethnologue) speaking them as a second language; the top five (English, Mandarin Chinese, Hindi, Spanish and French) have a combined total of just over 4 billion speakers, roughly half the global population (Eberhard et al., 2023). On the contrary, 97.2% of the total are spoken as a native language by only 12% of the global population. According to Ethnologue, 3,045 of the world's languages – more than 40% of the total – are

currently in danger of becoming extinct (Eberhard et al., 2023). Other recent estimates (Campbell and Belew, 2018) put the number of languages under threat at closer to 50%.

A language is classified as *extinct* by organisations like Ethnologue and UNESCO when it has no living speakers. This is something that has happened to many languages over the course of human history. Indeed, it is believed that most languages that have ever been spoken are now extinct (Crystal, 2002, 68–69). However, the past five centuries have seen a dramatic rise in the disappearance of the world's languages. Writing at the turn of the millennium, Daniel Nettle and Suzanne Romaine observed that Aboriginal Australian languages, of which there were over 250 before European contact, were "dying at a rate of one or more per year", while only around half of the pre-contact languages of the US survived, most of them "barely hanging on, possibly only a generation away from extinction" (Nettle and Romaine, 2000, 4–5). In South America, 25% of languages in the Tupian language family and 30% in the Arawakan family have stopped being spoken since 1970 (Loh and Harmon, 2014, 32).

The current decline in the world's languages arguably has its origins in European settler colonialism, with colonisers seizing land, displacing Indigenous peoples and imposing Western economic and cultural hegemony, with devastating effects on Indigenous languages and cultures. In the context of efforts to decolonise, narratives and terminology around linguistic and cultural "extinction" have been criticised as harmful to Indigenous peoples working to reclaim their languages and cultures (Amery, 2016; Muehlmann, 2017; Molina Vargas et al., 2020), with some preferring to refer to languages as *sleeping* or *no longer spoken*. We will use the term *extinction* in this paper while acknowledging the problematic nature of the term.

Returning to the context of *The Man Who Spoke Snakish*, where the hunter-gatherer narrator experiences displacement by farmers, it has been argued that at a global level, one of the main underlying causes of language death has been the transition of most human societies to agriculture over many millennia (Diamond, 1998; Diamond and Bellwood, 2003). According to this view, as all humans were hunter-gatherers living in relatively small groups until the Pleistocene, the gradual spread of sedentary, large-scale agricultural societies had a significant effect on the world's languages. This is because farmers, requiring more land to cultivate, encroached on land occupied by hunter-gatherers and drove them away, killed them or absorbed them into their increasingly centralised and hierarchical way of life through social networks, including marriage. The cultural expansion of agriculture was seen to be linked to the spread of the languages of the farmers at the expense of the languages of the hunter-gatherers, a process that has continued into the present day (Nettle and Romaine, 2000). Others view the transition to agriculture as much more patchy and experimental, and not tied in a causal way to the emergence of economic and social hierarchies (Graeber and Wengrow, 2021). The relationship between the pre/historic spread of agriculture, acculturation and colonialism has also been the subject of much debate. Jochim (2009) reviews evidence from recent and prehistoric agricultural expansion as colonisation and notes the experimental and uneven nature of the spread of farming. He notes that as Iban horticulturalists spread across Sarawak on the island of Borneo beginning in the sixteenth century, some hunter-gatherers fled the area in violent encounters while others acculturated (Jochim, 2009, 304). Gosden (2004) rethinks notions of colonialism, distinguishing between "middle ground" colonialism which involves a certain amount of co-construction of a shared culture (e.g. the Roman and Inca empires), and *Terra nullius* colonialism in which colonisers employ extreme violence and land displacement on colonised peoples, often exacerbated by the spread of contagious zoonotic diseases (e.g. settler colonisation by Russia; the British in Australia; the Spanish in Central America).

The link between agricultural land use, migration and linguistic diversity is articulated by Crystal (2002, 70–76), who notes that in some cases, agricultural land is outright stolen or invaded, often accompanied by the murder of Indigenous people, as has happened frequently in the Amazon rainforest. The loss of arable land – which can be a major factor in the displacement of a language's speakers and its eventual extinction – is frequently the result of overcultivation, overgrazing, cash-cropping or deforestation. Likewise, famine and drought, though clearly impacted by factors beyond human control, can also be caused or worsened by societal factors. Crystal identifies the Irish potato famine of the mid-nineteenth century as a contributing factor in the decline of the Irish language (Crystal, 2002, 71). Although the trigger cause of the famine was the arrival of potato blight in Ireland, the systems of landholding and single-crop dependency, both of which laid the groundwork for the blight to develop into a famine, were primarily the results of decisions by the British ruling classes; decisions by the ruling classes, such as landlords' evictions of Irish tenants, and inadequate famine relief from the British government, also worsened the catastrophe.

There are other, more direct ways in which colonialism leads to language decline. Crystal identifies a number of ways that languages may be lost through cultural assimilation. This may happen as a result of "demographic submersion", where new arrivals "swamp" the Indigenous culture with their language and material culture, as happened in the *Terra nullius* settler colonisation of North America and Australia (Crystal, 2002, 77). Religion has also played a role as a vector for linguistic assimilation and has a mixed legacy. While some Christian missionaries mastered and documented disappearing Indigenous languages, leaving valuable records for language reclamation later on (e.g. for Kaurna, Amery, 2016), others suppressed Indigenous languages. Jesuit missionaries forced Indigenous Brazilians onto Portuguese-speaking missions; and Catholic boarding schools in Canada punished First Nations children for speaking their languages. The spread of Christianity as an aspect of cultural assimilation is a key theme in *Snakish*.

Dominant cultures and governments have taken steps to deliberately suppress and eradicate languages. At one extreme, language suppression can take the form of outright extermination of speakers, as in the case of the Indigenous communities of El Salvador: in 1932, in response to a failed uprising, the government of Maximiliano Hernández Martínez launched a genocide that killed roughly 25,000 people from Indigenous Pipil communities. In addition to the speakers who were killed in this genocide, many native Pipil speakers who survived stopped speaking the language, as they feared that their speech would make them identifiable as "Indians", which could easily result in their death (Nettle and Romaine, 2000, 6). In other contexts, discriminatory language policies imposed by governments target children through educational systems to break the transmission of language from parents to children. Indigenous and tribal children were forced to attend monolingual boarding schools in the Soviet Union, India, Canada and the US. At these schools and throughout the British Empire, from Kenya to Wales, children were punished for speaking their own tongues (and often incentivised to tell on others who did, as in the case of the notorious Welsh Not) (Crystal, 2002, 84–85). Over generations, these steps lead to the stigmatisation of minoritised languages as "barbaric", "useless", "backward" and of lower status, which means that parents are less likely to transmit them to children, and the language dies out.

The insidious legacy of colonialism, driven by global capitalism, is implicated by Roche (2022) in what he calls the "necropolitics of language oppression". He links language death to a loss of speaker agency and autonomy, adopting Taff et al.'s (2018, p. 862) definition of language oppression as the "enforcement of language loss by physical, mental, social and spiritual coercion". Roche goes on to present evidence that links language oppression to a loss of well-being and the physical death of speakers; to lose one's language is to lose one's life. From the point of view of speakers, the death of a language is also a loss of cultural identity, so that "facing the loss of language or culture involves the same stages of grief that one experiences in the process of death and dying" (Dauenhauer and Dauenhauer, 1998, 71).

In this brief summary, we have discussed the historical context of the current extinction crisis in the world's languages. These include the historical spread of agriculture, colonisation, cultural assimilation, and stigmatisation of Indigenous languages. In the remainder of the paper, we will look at a work of fiction that centres its narrative around the lived experience of language death: *The Man Who Spoke Snakish* by Estonian author Andrus Kivirähk.

## The decline of Snakish

*Mees, kes teadis ussisõnu* is an Estonian novel by Andrus Kivirähk, first published in 2007 (Kivirähk, 2013). It was translated into English by Christopher Moseley and published in 2015 as *The Man Who Spoke Snakish* (Kivirähk, 2017). The novel is set in a fantastical reimagining of thirteenth-century Estonia during the Northern Crusades, when the land is being invaded and settled by Germanic Christians. Kivirähk's novel tells the story of Leemet, one of the last Estonians to continue living a hunter-gatherer lifestyle in the forest, which the rest of his people are gradually abandoning in favour of village life, agriculture and Christianity. Leemet is also one of the last people – by the end of the novel, the very last person – to speak Snakish, an ancient language that was taught to humans by snakes many generations ago and that gives speakers the power to control most animals. Leemet's people have lived in close relation to the non-human world for millennia, practicing their traditional spirituality in sacred forest groves, hunting animals for meat, telling myths and stories and embarking on complex relationships with members of non-human animal species, especially snakes, wolves and bears. The novel employs magical realism alongside elements of folklore, including from the Estonian epic *Kalevipoeg* (Kreutzwald, 2011), the Finnish *Kalevala* (Lönnrot, 1888) and Estonian folk tales such as "The Northern Frog" (Kreutzwald, 1985)and "The Tale of the Man Who Knew Snake-Words" (Parijõgi, 1977). The shamanic culture of Leemet's Estonians bears similarities to traditional Finno-Ugric culture, for example, worship in sacred groves (practiced to this day by the Mari people of the Russian Volga). In *Snakish*, the way of life of the "old times" and the Snakish language are both rapidly disappearing as hunter-gatherers leave the forest to settle in villages.

Some scholars, such as Niitra (2011) and Ehala (2007), have interpreted Snakish as a metaphor for the Estonian language. Indeed, Kivirähk himself has suggested in an interview that in a hundred years, the original Estonian version of his novel may be "full of snake words that no one can read" (Rooste, 2007, our translation), suggesting that he fears Estonian may follow the same path as his fictional Snakish language. Estonian is a member of the Uralic language family, which also includes Finnish and Hungarian. According to UNESCO (n.d.), most of the languages related to Estonian are extinct (Kamas, Ter Sámi), severely endangered (Ingrian, Votic) or endangered (Udmurt, Inari Sámi). Estonian has over a million speakers and is the national language of Estonia, but it has a reputation as being particularly difficult to learn as a second language, partly because of its status as a non-Indo-European language and partly because of its complex system of grammatical cases and phonological length distinctions which make it challenging even for speakers of related languages such as Finnish (Abdullah, 2015; Visit Estonia, 2022). Similarly, Snakish is a difficult language for humans to learn. His uncle, Vootele, warns Leemet that Snakish is "not easy, and that's why many people today can't be bothered with it" (28).[1] Leemet himself tells the reader that "Snakish words are not simple; the human ear can hardly catch all those hairline differences that distinguish one hiss from another, giving an entirely different meaning to what you say" (29), which could be a reference to the vowel and consonant length distinctions that make Estonian particularly difficult for non-native speakers.

The decline of Snakish in the novel, then, can be seen as a parallel for the difficulties that the Estonian language has faced over the centuries. The inhabitants of what is now Estonia were some of the last people in Europe to retain their pre-Christian spiritual traditions, triggering a series of Crusades into the area led by Germans, Danes and Swedes. Estonia (known for a time as Livonia) was then occupied by Sweden, Poland-Lithuania and Russia. In the twentieth century came the language's active suppression under Soviet rule. During this time, the Soviets deported tens of thousands of Estonians to work camps in Siberia and Kazakhstan, and Russian speakers were brought to live in Estonia. This was part of a larger "Russification" project by Josef Stalin to force members of the diverse ethnic groups living in the Soviet Union to assimilate to Russian language and culture. As a result of mass deportations and forced monolingual boarding schools for ethnic minority children, about 70 languages in the USSR became extinct (Rannut, 1995; Nettle and Romaine, 2000, 195–196; Moore, 2006, 17). In Estonia, Russian was the language of prestige and power, while many Estonian language texts (20,000 books and 5,000 volumes of periodicals, by one estimation) were destroyed (Avgerinos, 2006). In the post-Soviet era, from 1991 to the present day, Estonian has faced pressures from international languages like English. In the same interview where he speculates about his own work becoming unreadable "snake words", Kivirähk observes that "everyone who lives [in Estonia today] speaks English to each other. Or I-don't-know-what language" (Rooste, 2007, our translation). Given the historical context of Estonian and languages related to it, it is not surprising that Kivirähk and other speakers feel threatened by the cultural dominance of other languages, and this sense of insecurity, decline and impending extinction pervades the novel. German and Latin are the two languages that the villagers mention as being particularly useful, but these languages can be seen as a proxy for Russian, English and any number of more recent linguistic "invaders".

It should be noted that Kivirähk's novel does not overly idealise or romanticise the world of pre-Christian Estonia. As Kaljundi (2007) observes, the ancient forest sage Ülgas and his devoted follower Tambet are presented as just as ignorant as – and considerably more dangerous than – the Christian monks. Indeed, the novel satirises various icons of Estonian national identity, including "hardworking peasants" and "wise hedgehog[s]" (Niitra, 2011, 54; Howard, in press). Despite this, the Snakish language itself remains

---

[1]Unless otherwise stated, bracketed page numbers refer to the English translation of Kivirähk's novel (Kivirähk, 2017).

a potent and empowering force throughout the novel. When Leemet learns that some of the villagers share Ülgas and Tambet's belief in sprites, he laments how "dreadful" it is that the villagers believe in these "fairy-tale characters" while "den[ying] Snakish", the greatest wisdom that the old world of the forest could have taught them (209). Niitra writes that, "[w]hile the text sneers at other historical myths important for Estonians, snake words are the only thing to remain sacred. Obviously, snake words work as a metaphor for the native language" (Niitra, 2011, 57). Despite Kivirähk's often satirical and sarcastic approach towards other symbols of his nation's past, he presents the Estonian language as a powerful and important aspect of the people's identity.

*The Man Who Spoke Snakish* also overtly references the sociocultural effects of the Christian Crusades into the area. The decline of the Snakish language in the novel is largely driven by the movement of Estonian people from the forest to the nearby village, where Christian doctrine teaches that "[t]he snake is the right hand of Satan" (113). When Leemet visits the village and tries to tell the village elder Johannes about Snakish, Johannes declares that "[n]owhere on earth do they talk to snakes" and celebrates the fact that the Estonians are gradually forgetting the language, because "God doesn't want us to talk to snakes, for a snake is his enemy" (195). This resonates with, for example, the lived experience of Haida-speaking Alaska Natives forced to use English by institutions like the Church and schools in the twentieth century; evangelical preachers told the Haida that their traditional culture and language were "demonic" and that "God does not like Haida" (Dauenhauer and Dauenhauer, 1998, 64–65).

Instead, the villagers in Kivirähk's novel are keen to learn foreign languages, in particular German and Latin, as mentioned above. After dismissing Snakish as a useless language, the village boys try to impress upon Leemet the value of European languages:

> "It would be a different matter if you knew Latin well," said Andreas. "Then you'd sing hymns and you'd get all the women into bed." […]
>
> "German is important too," added Jaakop. "That's what the knights speak. If you understand German, some knight might take you as his servant" (217).

With the second example, Kivirähk is making a wry comment about how speaking the language of the colonisers can also make you their servant. The perception that some languages are more "useful" than others is one of the main drivers of language shift, where communities of speakers may willingly adopt a different language from that of their parents in order to benefit socially or economically (Fishman, 1991). Nettle and Romaine distinguish between "metropolitan" and "peripheral" languages, explaining that:

> Metropolitan languages are associated with a dominant economic or social class […] [and] with economically leading central places […] Peripheral languages, by contrast, are restricted to economically less developed areas, and also to a smaller range of economic roles and functions (Nettle and Romaine, 2000, 128).

Within the context of Kivirähk's novel, Latin and German can be seen as metropolitan languages, while Snakish is a peripheral language. Latin and German are the languages of the Catholic Church and knights, respectively, and both hold high prestige among the villagers. Latin is associated with "the holy city of Rome", a centre that embodies, in Johannes's words, "the might of the world" (197). In contrast, Snakish is only spoken in the "economically less developed" forest, where food is obtained by hunting and gathering, rather than farming. Even then, although Snakish is used for communication between humans and other animals, it is rarely used for conversations between humans.

Although the decline of Snakish is driven partly by the Estonians' choice to move to the village, there are also instances of clear, physical violence enacted by the villagers against the language's speakers. When Leemet travels to the village accompanied by his friend Ints the adder, Johannes attempts to kill the serpent, and later, the villagers burn a whole nest of snakes in which Leemet's mother had also been hibernating. In a way, this can be seen as a genocide of Snakish speakers (both human and non-human), motivated by the villagers' zealous hatred of snakes. The fact that humans and snakes are burned together in this attack also links to the fact that Snakish, in some ways, dissolves the boundaries between humans and non-humans. This matter is the focus of the following section.

## Nature and culture

One of the main uses of Snakish for Leemet and his family is for hunting, since Snakish words can be used to force wild animals to submit to the will of the speaker. In a prologue set after the other events of the novel, Leemet describes a deer "down on his knees and submissively offer[ing] me his neck, just as in the old days, when we used to get our food this way – by calling the deer to be killed" (2). Snakish is even used to effectively domesticate wolves, as Leemet explains:

> [Wolves] will serve humans, carrying them on their backs and allowing themselves to be milked – though only under the influence of the Snakish words. A wolf really is a fairly dangerous domestic animal, but since there is no tastier milk to be had from anyone in the forest, one reconciles oneself to its sullenness, especially as the Snakish words render it as meek as a titmouse (16).

Part of the Snakish language's value for the forest-dwellers is as a tool for controlling the animals of the forest to obtain food (e.g. meat and milk). Peiker notes that, for most animals other than snakes themselves, "Snakish simply works as a magic spell the orders in which they must – quietly – obey" (Peiker, 2016, 124). Using Snakish, humans can be masters of non-human animals, in contrast to the languages like German, which, as discussed above, has the ability to make the Estonians servants to foreigners.

However, the Snakish language in Kivirähk's novel also represents a blurring of the distinction between nature and culture: it is an element of human culture (a language) that has been taught to humans by non-human nature (snakes). Snakish is a reminder that humans are a part of nature and that even attributes that we identify as distinctly human (that is, culture) arise from nature. Leemet's hunter-gatherer culture is inextricably linked with the Estonian forest ecosystem, very much in keeping with Loh and Harmon's (2014) model of "biocultural diversity", which stresses the interrelationship between ecology, culture and language. Although it facilitates violence against non-human animals through hunting and domestication, the Snakish language also helps to situate humans as animals within the wider ("natural") world. With the power that it grants humans, Snakish transforms the forest into a safe – even tame – location; for those ignorant of the language, however, the forest is a wild and dangerous place.

Indeed, the Snakish language in Kivirähk's novel blurs the very boundary between humans and non-humans. Vootele vows to "teach Leemet the Snakish words so well that he won't know anymore whether he's a human or a snake" (31). This ability of Snakish to make human speakers physically more snakelike is exemplified by Leemet's grandfather, who appears after having been believed to be dead for many years. Still a fluent Snakish speaker, he has been living on an island without legs, crawling

around "like some hairy adder" (250) and biting anyone (apart from Leemet and his wife, Hiie) who arrives on the island and killing them with his "blackened but still sharp fangs" (249) – a trait which he is disappointed to learn has not been passed on to his descendants. The decline of the Snakish language among humans is associated with humans become physically more differentiated from snakes, losing their "ancestral" fangs and wildness (249) and moving away from the forest. Although there is no explicit link between language and ecosystem in the novel, as the Snakish language dies out, humans become more alienated from the forest and non-human life, until even Leemet is treating the other animals of the forest with disdain.

Sõrmus identifies Leemet's grandfather as one of many "instances of naturalcultural hybridity" which blur the boundaries between non-human nature and human culture in Kivirähk's novel (Sõrmus, 2015, 49). Another such example is the relationships between humans and bears in the novel: Leemet's older sister Salme falls in love with a bear called Mõmmi, whom she later marries, and Leemet tells the reader that women falling for bears is a "familiar story" among the forest-dwellers (14). The nature-culture boundary is also transgressed by the mysterious character Meeme, an old forest-dweller who appears to be slowly decaying over the course of the novel, "sinking more and more under the sod" until he himself becomes "human sod" (179) and "dissolve[s] into nature […] his eyes […] like dewdrops" (433). While Leemet and his family members transgress the boundary between humans and non-human animals (snakes and bears), Meeme transgresses – indeed, dissolves through – the boundary between humans and the non-animal elements of the forest: the earth, moss and dew. At the same time, Meeme's slow disintegration, as well as the lingering stench of death in Leemet's nostrils after he is trapped underground with his uncle's corpse, could be seen as a comment on the emotional experience of witnessing one's culture and language decline on a path towards extinction.

The forest-dwellers' way of life is contrasted with that of the villagers. While the forest-dwellers continue to converse with snakes and bears and live surrounded by the animals, plants and fungi of the forest, the villagers view nature as a foreign, dangerous entity that must be tamed: for Johannes, the forest is a "dark thicket" (24) where "beasts of prey walk about and Satan rules" (193). The villagers spend their days ploughing and harvesting the fields, forcing the land to produce what they want, rather than taking what nature provides, as the forest-dwellers do. Their relations with non-human beings are not necessarily more violent than those of the forest-dwellers: while the forest-dwellers hunt to their hearts' content, the villagers only eat meat on holy days, instead living mostly off bread and porridge. However, the villagers view themselves as both separate from and superior to non-human nature, and their rejection of Snakish epitomises this anthropocentric approach. After celebrating the coming extinction of Snakish (see above), Johannes changes tack and declares that the language actually does not exist at all, because "God hasn't given snakes the power of speech" (198). Discussing this passage, Sõrmus notes that "the belief in nature's muteness originates from the villagers' anthropocentric stance: nature having not been *given* a speaking status, so that it is relegated to the place of silence" (Sõrmus, 2014, 184, emphasis in original). Rather than human culture deriving from nature, for the villagers, the *distinction between* humans and nature derives from God, who is at the apex of the natural-social-divine hierarchy that echoes the Great Chain of Being.

However, while Snakish defies the Christians' hierarchy, the magical language also constructs its own Chain of Being within the forest, where "[t]he hierarchy of all the creatures is based upon and expressed through Snakish" (Peiker, 2016, 124). Snakes and Snakish-speaking humans are the supreme beings in this system, while creatures that have no understanding of the language, such as ants, are "just tiny specks of dirt with legs, not even worth noticing" (148). This language-based hierarchy may have problematic implications for contemporary Estonian language politics, where, in some cases, knowledge of the Estonian language is a requirement for citizenship (Howard, in press). In particular, the association between language and the local environment (as embodied in Snakish) implies that speakers of a native language – whether the fictional Snakish or the real Estonian – have a greater connection to their home than those who live there but speak other languages (German, Russian, English, etc.). The eco-linguistic nationalism that may arise from such a reading, however, seems to be strongly derided by Kivirähk himself, who has expressed fears over nationalist violence and aggression in Estonia and throughout Europe (Rooste, 2007), and who has even stated that "everything good about Estonia has been taken from the Germans" (Vaino, 2021). As discussed above, Kivirähk is happy to satirise national symbols and narratives, and even his celebration of a national language hints at a violent and oppressive potential.

## Conclusion

*The Man Who Spoke Snakish* is set in Medieval Estonia, but the viewpoint of the protagonist as a speaker of a vanishing language from a vanishing culture resonates with the lived experience of millions of people who have lost lifeways and livelihoods to colonialism and cultural assimilation. Like many Indigenous languages, the fictitious language Snakish allows its speakers to integrate fully into the natural world and to form complex interdependent relationships with non-human animals. This web of nature, culture and language is destroyed by a society that is anthropocentric, ecologically destructive and socially hierarchical, and which views nature as something to exploit or fear. The novel explores the emotions of grief and loss for both a culture and an ecosystem heading for extinction. It also, through fantastical allegory and allusion, explores the relationship between nature and culture, and the ways in which colonialist policies and mindsets threaten both.

**Open peer review.** To view the open peer review materials for this article, please visit http://doi.org/10.1017/ext.2024.3.

**Financial support.** This work was supported by the Leverhulme-funded Extinction Studies Doctoral Training Programme at the University of Leeds.

**Competing interest.** The authors declare no competing interests exist.

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
