## [Editor Report · Recommendation: Snake words in Estonia: Language, nature and extinction in Andrus Kivirähk’s *The Man Who Spoke Snakish*
— R0/PR2]

Dear Alfie and Diana,

We are happy to inform you that your article has been accepted for publication in this special issue of Prisms: Extinctions. As you can see below, however, the reviewers have recommended major revisions, both with respect to the historical background and interpretation of the novel, with both reviewers highlighing the inappropriateness of treating it as a historical novel, let alone a historical resource. We hope that you will be ready, willing and able to revise and resubmit along the lines suggested here. Please do not hesitate to contact us if you have any queries about how to respond to these recommendations.

Reviewer 1:

The paper discusses problems of language extinction and cultural assimilation with reference to Andrus Kivirähk’s novel ‘The Man Who Spoke Snakish’. The paper is well written, logically argued and based on a wealth of sources ranging from biocultural diversity to reviews of Kivirähk’s novel. One problematic issue seems to be the interpretation of Kivirähk’s work as a historical novel. Kivirähk has always been a highly creative and parodistic writer. His works have a strong satirical component. It seems that the authors' critical attention to the extinction of languages and cultures leads them to interpret Kivirähk’s work too realistically. In the chapter “Nature and Culture”, some of Kivirähk’s satirical mode is visible (such as “milking the wolves”, the bear named Mõmmi who is Salme’s companion (Mõmmi is a common nickname for teddy bear in Estonian), etc.), but not commented on by the authors. Kivirähk seems to exaggerate and ridicule most historical narratives (both Western and non-Western). The difference in Kivirähk’s sympathy may lie between narratives that introduce rigidity and seriousness into the world, and those that allow for playfulness and imagination (cf. Joseph Meeker’s “comic mode”). A more nuanced interpretation of Kivirähk’s novel could be found, for example, in the magazine Vikerkaar (Linda Kaljundi’s “Aga ükskord ei alga aega”, https://www.vikerkaar.ee/archives/11107). The paper would benefit from more developed conclusions that could integrate the analysis of Kivirähk’s novel and the issue of language extinction more deeply. Despite these comments, the paper is a remarkable piece of research that uses the example of the novel, written in a very small language, to illustrate the serious problem of cultural homogenisation and language extinction.

Reviewer 2:

This article addresses an important and complex question regarding the relationship between biodiversity loss and language loss in contexts of colonisation in association with a discussion of a contemporary Estonian novel, which narrativizes this conjunction in the mode of historical fantasy fiction. The article is generally well-written, but the research is a little thin in places, and there are inadequacies in the argumentation as well as in the contextualisation and analysis of the novel. There are two main aspects in which substantive revision is required.

Firstly, it should be acknowledged that the possible association between loss of biological and linguistic diversity has been discussed for some time within the field of ecolinguistics (see e.g. several chapters in Fill and Mühlhäusler, The Ecolinguistics Reader: Language, Ecology, and Environment, 2001). Empirical research nonetheless remains relatively sparse, and the nature of the association is debated. Does one cause the other, or are they correlated with underlying factors, and if so, under what particular conditions of colonisation do they occur together? A clearer distinction also needs to be made between colonial expansion that engenders a shift from hunter-gatherer (or e.g. gardening or firestick farming) societies to agrarian societies, and cases of colonisation (such as that of the English in Ireland), where an existing agrarian regime is displaced by other forms of landuse. With respect to the longer history of agrarianism, it should be noted that the process was geographically and temporally patchy, rather than all at once at the end of the Pleistocene. Here, I would recommend augmenting or replacing the rather pop science account of Jared Diamond by more serious archeological research e.g. Jochim, Michael. “The process of agricultural colonization.” Journal of anthropological research 65.2 (2009): 299-310 and references therein, esp. Gosden, Chris. Archaeology and colonialism: cultural contact from 5000 BC to the present. Vol. 2. 2004. It is noteworthy that Gosden argues that modern colonialism, by giving rise to settler societies, is historically unusual. With respect to the specific case of Estonia, what evidence is there that the shift to farming following the Northern Crusades led to a loss of linguistic and/or biological diversity? With respect to the former, at least, the article seems to suggest that this was mainly a facet of (atheistic) Soviet domination, rather than Christianisation. Did this more recent experience of imperialism also contribute to steeper biodiversity loss?

Secondly, the novel needs to be more carefully contextualised and analysed with greater critical rigour. As discussed here, it appears to present a highly reductive black and white contrast between indigenous eco-pagans and eco-phobic Christian invaders. This is an extreme historical distortion, considering that the first major wave of deforestation in Europe and North Africa was the work of the polytheistic Roman empire, whereas there is a long tradition, going back to the 4th C of Christian hermits seeking closeness to God in deserts and woodlands, with stories of their companionate relations with wild animals, as well as monastic communities restoring and caring for woodlands, soils and streams, which afforded spiritual as well as physical nourishment in accordance with a view of nature as God’s good creation (see e.g. Torvend, Monastic Ecological Wisdom: A Living Tradition, 2023). The armed adventurers who took part in the Northern Crusades might well have been of a very different disposition, and it is certainly true that in more recent settler colonial experience, the type of Christianity imposed upon Indigenous peoples had become infected by the geopolitical and economic interests of mercantile and extractivist capitalism. However, the novel, as discussed here, presents a gross caricature of mediaeval Christianity, along with what is almost certainly a romanticised view of pre-Christian Estonian society. That is ok, in so far as we should not judge works of fiction by their historical accuracy. But it raises the question of the ecopolitical context and implications of the novel. In my view, it is profoundly unhelpful with respect to contemporary decolonial ecopolitics, where, e.g. the Catholic church, through grassroots organisations as well as papal leadership, is actively seeking to defend forest-dwelling Indigenous peoples of the Amazon against neocolonial capitalist extractivism and genocide (see e.g. http://secretariat.synod.va/content/sinodoamazonico/en/the-pan-amazonian-region/repam.html). On the other hand, the novel’s celebration of Indigenous Nordic paganism might well be appealing to some eco-nationalists, or even eco-fascists. This is not to suggest for a moment that this right-wing politics would be endorsed by either the novelist or the author of the article. These are simply exemplary of aspects of the contemporary context in which the persuasive force of a novel such as this needs to be critically reflected.

---

## [Editor Report · Recommendation: Snake words in Estonia: Language, nature and extinction in Andrus Kivirähk’s *The Man Who Spoke Snakish*
— R1/PR5]

I recommend that this move to acceptance. I am satisfied with the revisions done to address the reviewer comments.